# DSpar : An Embarrassingly Simple Strategy for Efficient GNN Training and Inference via Degree-Based Sparsification

**Zirui Liu**[1], **Kaixiong Zhou**[1], **Zhimeng Jiang**[2], **Li Li**[3], **Rui Chen**[3], **Soo-Hyun Choi**[3], **Xia Hu**[1]

*{Zirui.Liu, Kaixiong.Zhou, Xia.Hu}@rice.edu, {zhimengj}@tamu.edu, {li.li1, rui.chen1, sh9.choi}@samsung.com*
[1]*Rice University,* [2]*Texas A&M University,* [3]*Samsung Electronics America*

**Reviewed on OpenReview:** *https://openreview.net/forum?id=SaVEXFuozg*

## Abstract

Running graph neural networks (GNNs) on large graphs suffers from notoriously inefficiency. This is attributed to the sparse graph-based operations, which is hard to be accelerated by community hardware, e.g., GPUs and CPUs. One potential solution is to "sketch" the original graph by removing unimportant edges, then both the training and inference processes are executed on the sparsified graph with improved efficiency. Traditional graph sparsification work calculates the edge importance score, i.e., effective resistance, from graph topology with theoretical guarantee. However, estimating effective resistance is even more expensive than training GNNs itself. Later, learning-based sparsification methods propose to learn the edge importance from data, but with significant overhead due to the extra learning process. Thus, both of them introduce significant ahead-of-training overhead. In this paper, we experimentally and theoretically prove that effective resistance can be approximated using only the node degree information and achieve similar node presentations on graphs with/without sparsification. Based on this finding, we propose DSpar to sparsify the graph once before training based on only the node degree information with negligible ahead-of-training overhead. In practice, in the training phase, DSpar is $1.1 \sim 5.9\times$ faster than the baselines on different datasets and models with almost no accuracy drop. For the inference phase, DSpar reduces $30\% \sim 90\%$ latency. The code is available at `https://github.com/warai-0toko/DSpar_tmlr`.

## 1 Introduction

Graph neural networks (GNNs) have achieved great success in representation learning on graphs from various domains, including social networks (Hamilton et al., 2017; Liu et al., 2023a; Jiang et al., 2022), biology (Hu et al., 2020), and recommendation system (Ying et al., 2018; Liu et al., 2023b; Zha et al., 2023; Wang et al., 2020). Despite their effectiveness, GNNs are notoriously known for being inefficient. Specifically, GNNs are characterized by their alternating sequence of aggregation and update phases. During the aggregation phase, each node gathers information from its neighboring nodes in a layer-by-layer manner, using **sparse matrix-based operations** as described in Fey & Lenssen (2019); Wang et al. (2019). Then, in the update phase, each node updates its representation based on the aggregated messages via **dense matrix-based operations** Fey & Lenssen (2019); Wang et al. (2019). In Figure 1, `SpMM` and `MatMul` are the sparse and dense operations in the aggregation and update phases, respectively. Our profiling results indicate that the aggregation phase may take up to 90% of the total training time, and a similar pattern can be observed in the inference process.

The execution time of sparse operation is proportional to the number of edges in the graph. So intuitively, one straightforward way to accelerate the process is to sparsify the graph by removing unimportant edges. Thus, previous work tries to produce a "sketch" of the input graph by removing unimportant edges once before training, then using the sparsified graph for both the training and inference process. These works can be

roughly divided into two streams of works. **The first stream of work** proposes to remove edges according to their **theoretical importance score**, which is calculated based on the graph topology (Spielman & Srivastava, 2011). Albeit the good theoretical properties, obtaining the theoretical importance score (i.e., effective resistance) is even more expensive than training GNNs itself. **The Second stream of work** tries to **learn the edge importance from data** (Zheng et al., 2020a; Chen et al., 2021; Li et al., 2020). However, learning to drop edges introduces another training process for identifying redundant edges, which introduces significant ahead-of-training overhead. Thus up to our knowledge, no prior discussion was placed on how to accurately sparsify the graph with little or no time overhead for fast GNN training and/or inference. In view of such, this paper raises the question:

*Can we sparsify the graph once before training with little overhead, while achieving similar model performance with less training and inference time?*

This paper makes an attempt in providing a positive answer to the above question. We experimentally and theoretically prove that the traditional edge importance score, i.e., effective resistance, can be efficiently approximated by using only the node degree information. Based on our theoretical analysis, we propose Degree-based Sparsification (DSpar ). Specifically, We first down-sampling edges based on only the node degree before training. Then we use the sparsified graph for both training and inference. As a result, the training and inference processes are both accelerated since computations are executed on a sparsified graph. We theoretically show that the node embeddings learned on the graph sparsified by DSpar are good approximations of those learned on the original graphs. Our main contributions are outlined below:

- We design DSpar , a highly-efficient algorithm to sub-sample the edges based on only the degree information before training. We theoretically prove that GNNs could learn expressive node representations on graphs sparsified by DSpar .

- DSpar sparsifies the graph once before training, and the sparsified graph can be used for both training and inference with improved efficiency. DSpar allows to run GNNs faster in wall-clock time. For the training phase, DSpar achieves $1.1 \sim 5.9\times$ faster than baseline with almost no accuracy drop. For the inference phase, DSpar reduces $30\% \sim 90\%$ latency compared to the baseline.

- We implement DSpar as a ready-to-use extension for Pytorch Geometric and Pytorch, which supports parallel sampling of edges from large graphs (Appendix A).

## 2  Preliminary Analysis

Let $\mathcal{G} = (\mathcal{V}, \mathcal{E})$ be an undirected graph with $\mathcal{V} = (v_1, \cdots, v_{|\mathcal{V}|})$ and $\mathcal{E} = (e_1, \cdots, e_{|\mathcal{E}|})$ being the set of nodes and edges, respectively. Let $\boldsymbol{X} \in \mathbb{R}^{|\mathcal{V}| \times d}$ be the node feature matrix. $\boldsymbol{A} \in \mathbb{R}^{|\mathcal{V}| \times |\mathcal{V}|}$ is the graph adjacency matrix, where $\boldsymbol{A}_{i,j} = 1$ if $(v_i, v_j) \in \mathcal{E}$ else $\boldsymbol{A}_{i,j} = 0$. $\tilde{\boldsymbol{A}} = \tilde{\boldsymbol{D}}^{-\frac{1}{2}}(\boldsymbol{A} + \boldsymbol{I})\tilde{\boldsymbol{D}}^{-\frac{1}{2}}$ is the normalized adjacency matrix, where $\tilde{\boldsymbol{D}}$ is the degree matrix of $\boldsymbol{A} + \boldsymbol{I}$. GNNs recursively update the embedding of a node by aggregating embeddings of its neighbors. For example, the forward pass of the $l^{\text{th}}$ Graph Convolutional Network (GCN) layer Kipf & Welling (2017) can be defined as:

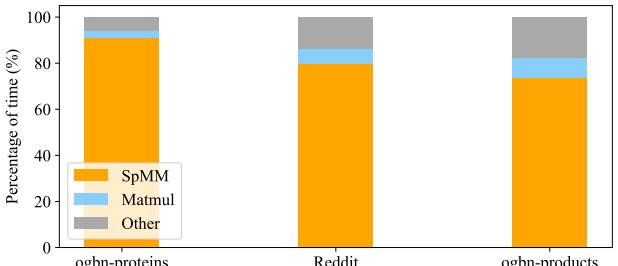

Figure 1: The time profiling of a two-layer GCNs on different datasets. `SpMM` in the aggregation phase may take $70\% \sim 90\%$ of the total time.

$$\boldsymbol{H}^{(l+1)} = \text{ReLU}(\tilde{\boldsymbol{A}}\boldsymbol{H}^{(l)}\boldsymbol{\Theta}^{(l)}), \tag{1}$$

where $\boldsymbol{H}^{(l)} \in \mathbb{R}^{|\mathcal{V}| \times d}$ is the node embedding matrix consisting of node embedding $\boldsymbol{h}_v^{(l)}$ for all $v \in \mathcal{V}$ at the $l^{\text{th}}$ layer and $\boldsymbol{H}^{(0)} = \boldsymbol{X}$. $\boldsymbol{\Theta}^{(l)}$ is the weight matrix of the $l^{\text{th}}$ GCN layer. In practice, $\tilde{\boldsymbol{A}}$ is often stored in the sparse matrix format, e.g., compressed sparse row (CSR) Fey & Lenssen (2019). From the implementation aspect, the computation of Equation (1) can be described as:

$$\boldsymbol{H}^{(l+1)} = \mathrm{ReLU}\bigg( \mathtt{SpMM}\Big( \tilde{\boldsymbol{A}}, \mathtt{MatMul}(\boldsymbol{H}^{(l)}, \boldsymbol{\Theta}^{(l)}) \Big) \bigg), \tag{2a}$$

where $\mathtt{SpMM}(\cdot, \cdot)$ is Sparse-Dense Matrix Multiplication and $\mathtt{MatMul}(\cdot, \cdot)$ is the Dense Matrix Multiplication. Unlike normal dense matrix, the elements is randomly distributed in the sparse matrix. Thus sparse operations, such as $\mathtt{SpMM}$ , have many random memory accesses and are much slower than the dense counter-part Han et al. (2016). To get a sense of the scale, we show in Figure 1 that for GCNs, $\mathtt{SpMM}$ may take roughly $70\% \sim 90\%$ of the total training time.

## 3 Fast Graph Sparsification

As we analyzed, the sparse operations are the main efficiency bottleneck for running GNNs. For sparse operations, the computation is only executed on non-zero entries. To improve the efficiency, we propose to reduce the number of non-zero entries in the adjacency matrix by removing unimportant edges.

### 3.1 Sampling-base graph sparsification

To improve the training efficiency while minimizing the impact of compression, we sparsify the graphs by removing unimportant edges. **We sparsify the graph once before training. Thus, the training and inference processes are both accelerated since the computation is done on sparsified graphs.** Below we introduce how to sparsify the graph in an unbiased and efficient way.

---

**Algorithm 1:** Sampling-based Graph Sparsification Spielman & Srivastava (2011)

**Input:** $\mathcal{G} = (\mathcal{V}, \mathcal{E})$, sampling probability $\{p_e\}_{e \in \mathcal{E}}$, number of samples to draw $Q$.
**Output:** the sparsified weighted graph $\mathcal{G}' = (\mathcal{V}, \mathcal{E}')$ with edge weights $\{w_e\}_{e \in \mathcal{E}'}$

1   $\mathcal{E}' \leftarrow \{\}$
2   **for** $j = 1, \cdots, Q$ **do**
3     Sample an edge $e \sim \mathcal{E}$ with replacement according to $p_e$
4     **if** $e \notin \mathcal{E}'$ **then**
5       Add $e$ to $\mathcal{E}'$ with weight $w_e = \frac{A_e}{Q p_e}$
6     **end**
7     **else**
8       $w_e \leftarrow w_e + \frac{A_e}{Q p_e}$.
9     **end**
10 **end**
11 **return** $\mathcal{G}' = (\mathcal{V}, \mathcal{E}')$ with edge weights $\{w_e\}_{e \in \mathcal{E}'}$

---

The process of graph sparsification is shown in Algorithm 1. First, given a graph $\mathcal{G} = (\mathcal{V}, \mathcal{E})$, for each edge $e = (v_i, v_j) \in \mathcal{E}$, we need to decide the probability $p_e$ that it will be sampled ($\sum_{e \in \mathcal{E}} p_e = 1$). Then we need to decide the number of samples to draw $Q$, which implicitly controls the sparsity of the sparsified graph $\mathcal{G}'$. The sparsified adjacency matrix $\boldsymbol{A}' \in \mathbb{R}^{|\mathcal{V}| \times |\mathcal{V}|}$ of $\mathcal{G}'$ can be constructed as $\boldsymbol{A}'_{i,j} = w_e$ (line 5 and line 8) if $e = (v_i, v_j) \in \mathcal{E}'$ else $\boldsymbol{A}'_{i,j} = 0$. We note that $\boldsymbol{A}'$ is an unbiased estimation of $\boldsymbol{A}$, i.e., $\mathbb{E}[\boldsymbol{A}'] = \boldsymbol{A}$. To see this,

$$\mathbb{E}[\boldsymbol{A}']_{i,j} = \mathbb{E}[\sum_{k=1}^{Q} \mathbb{1}_e \frac{\boldsymbol{A}_{i,j}}{Q p_e}] = \sum_{k=1}^{Q} \mathbb{E}[\mathbb{1}_e] \frac{\boldsymbol{A}_{i,j}}{Q p_e} = \boldsymbol{A}_{i,j},$$

where $\mathbb{1}_e$ represents the event that the edge $e = (v_i, v_j)$ being sampled and we have $\mathbb{E}[\mathbb{1}_e] = p_e$.

It is straight-forward to see that $|\mathcal{E}'| = \mathcal{O}(Q)$. Intuitively, the approximation error will diminish if $Q$ approaches infinity, resulting in a denser $\mathcal{G}'$. The challenge is how to choose a good $\{p_e\}_{e \in \mathcal{E}}$ such that we can set $Q$ as small as possible under the given error.

In theory, Spielman & Srivastava (2011) shows that $\mathcal{G}'$ is an accurate approximation of the input graph $\mathcal{G}$ if we set $p_e$ in proportional to the effective resistance $R_e$ for each edge $e$. In spectral graph theory, the effective resistance $R_e$ is often used as the distance measure between two nodes by encoding the global topology of the graph (e.g., the cluster structure) Lovász (1993). Specifically, for each edge $e = (u, v)$, the effective resistance $R_e$ is defined as

$$R_e = (\mathcal{X}_u - \mathcal{X}_v)^\top \mathcal{L}^+ (\mathcal{X}_u - \mathcal{X}_v), \tag{3}$$

where $\mathcal{L} = \boldsymbol{I} - \boldsymbol{D}^{-\frac{1}{2}} \boldsymbol{A} \boldsymbol{D}^{-\frac{1}{2}}$ is the normalized graph laplacian matrix and $\mathcal{L}^+$ is the psudo-inverse of $\mathcal{L}$. $\mathcal{X}_u \in \mathbb{R}^{|\mathcal{V}|}$ is the elementary unit vector with a coordinate 1 at position $u$. When the sampling probability $p_e$ is in proportional to $R_e$, Spielman & Srivastava (2011) shows that the sample complexity of $Q$ is $\mathcal{O}(\frac{|\mathcal{V}| \log |\mathcal{V}|}{\epsilon^2})$, where $\epsilon$ is a constant which controls the approximation error.

Despite the good theoretical properties, estimating the effective resistance $R_e$ is non-trivial because (1) it requires approximating the pseudoinverse of the graph laplacian matrix, which is very time consuming for large graphs. (2) it is hard to parallelize the calculation for each edge $e$.

Albeit Algorithm 1 is well-established, up to our knowledge, there is no previous work investigating its usage for GNNs. This is because even estimating $R_e$ is extremely time consuming, which counteracts the acceleration effects of graph sparsification.

In the next subsection, we will discuss how to approximate the effective resistance efficiently.

## 3.2 Efficiently approximating effective resistance

Here we introduce how we approximate the effective resistance of each edge using only its local information (e.g., the node degree). Specifically, the following Theorem shows that for any edge $e = (u, v)$, its effective resistance $R_e$ is bounded by $\frac{1}{d_u} + \frac{1}{d_v}$, which can be easily calculated since it only involves the node degrees.

**Theorem 1** (Corollary 3.3 in Lovász (1993)). *For all $e = (u, v) \in \mathcal{E}$, we have $\frac{1}{2}(\frac{1}{d_u} + \frac{1}{d_v}) \leq R_e \leq \frac{1}{\alpha}(\frac{1}{d_u} + \frac{1}{d_v})$, where $\alpha$ ($\alpha \leq 2$) is the smallest non-zero eigenvalue of $\mathcal{L} = \boldsymbol{I} - \boldsymbol{D}^{-\frac{1}{2}} A \boldsymbol{D}^{-\frac{1}{2}}$.*

In Spectral graph theory, $\alpha$ indicates the connectivity of a graph and the bound is tight for well-connected graphs. Informally, the intuition is that for large graphs with cluster structures, the random walk is hard to escape from the local cluster since most of the edges are pointing to nodes within the same cluster. Thus, the local information dominates the information flow on the whole graph. From Theorem 1, instead of setting $p_e \propto R_e$, we propose to set $p'_e \propto \frac{1}{d_u} + \frac{1}{d_v}$ for edge $e = (u, v)$.

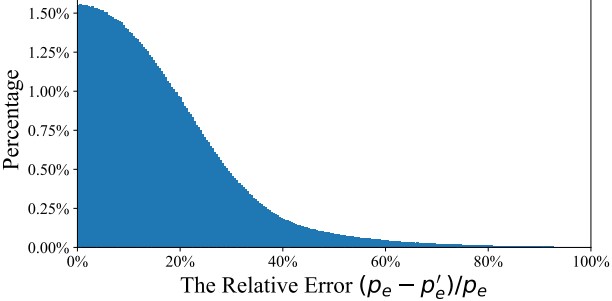

Figure 2: The histogram of the distribution of $\frac{p_e - p'_e}{p_e}$ for all edges in Reddit dataset.

**Experimental Analysis**. In Figure 2, we display the distribution of $\frac{|p_e - p'_e|}{p_e}$ for Reddit dataset, where $p_e \propto R_e$ and $p'_e \propto \frac{1}{d_u} + \frac{1}{d_v}$. We note that obtaining exact $R_e$ is impractical on large graphs since it is very time consuming. To get a sense, even the previous state-of-the-art approximation method still need **248 seconds** for estimating $R_e$. In contrast, calculating $\frac{1}{d_u} + \frac{1}{d_v}$ for all edge $e = (u, v)$ consumes only **0.6 second on the same hardware**. Experimentally, by setting $p_e \propto \frac{1}{d_u} + \frac{1}{d_v}$ for each edge $e = (u, v)$, applying Algorithm 1 to GNNs has negligible effects on the model accuracy (see Table 1), but significantly accelerates both the training and inference processes.

**Theoretical Analysis**. Here we theoretically analyze why our degree-based sampling prorduces a good approximation of the original graph. Formally, we have the following Theorem:

**Theorem 2** (Proof in Appendix B). *Given an input graph $\mathcal{G} = (\mathcal{V}, \mathcal{E})$, let $\boldsymbol{A}$ be the associated adjacency matrix and $\alpha$ be the smallest non-zero eigenvalue of $\mathcal{L}$. Given an error parameter $\epsilon$, If we set $Q = \mathcal{O}(\frac{|\mathcal{V}| \log |\mathcal{V}|}{\epsilon^2})$*

and for each edge $e = (u, v)$, we set $p_e \propto \frac{1}{d_u} + \frac{1}{d_v}$, Algorithm 1 produces a sparsified graph $\mathcal{G}' = (\mathcal{V}', \mathcal{E}')$ with $\boldsymbol{A}'$, for any vector $\boldsymbol{x} \in \mathbb{R}^{|\mathcal{V}|}$, we have

$$(1 - \frac{\epsilon}{\alpha}) \sum_{(u,v) \in \mathcal{E}} (x_u - x_v)^2 A_{u,v} \leq \sum_{(u,v) \in \mathcal{E}'} (x_u - x_v)^2 A'_{u,v} \leq (1 + \frac{\epsilon}{\alpha}) \sum_{(u,v) \in \mathcal{E}} (x_u - x_v)^2 A_{u,v}. \tag{4}$$

Informally, Equation (4) suggests that the degree-based sampling preserves the graph spectral information, i.e., the eigenvalues of the normalized Laplacian matrix. Specifically, let $\lambda_1 \geq \lambda_2 \cdots \lambda_{|\mathcal{V}|} = 0$ be the eigenvalue of $\mathcal{L}$ associated with $\mathcal{G}$, and $\lambda'_1 \geq \lambda'_2 \cdots \lambda'_{|\mathcal{V}|} = 0$ be the eigenvalues of $\mathcal{L}'$ associated with the sparsified $\mathcal{G}'$ given by Algorithm 1, we have:

$$(1 - \frac{\epsilon}{\alpha})\lambda_i \leq \lambda'_i \leq (1 + \frac{\epsilon}{\alpha})\lambda_i. \tag{5}$$

Here we defer the mathematical details of Equation (5) to Appendix B. According to the graph theory, small (large) eigenvalues indicate the global clustering (local smoothness) structure of the graphs (Zhang et al., 2019). Thus if the eigenvalues are similar, then most of the graph properties will be preserved, e.g., cluster structures and Cheeger constant (West et al., 2001). **Later we experimentally show that both the largest and smallest eigenvalues are persevered by DSpar (Section 5.2.2).** In the next Section, we show that preserving spectral information is crucial for maintaining the quality of node representations.

### 3.3 Why degree-based sparsification works for GNNs?

Here we analyze why the sparsification works for GNNs theoretically. High-levelly speaking, the reason there is a term "Convolutional" in the name of GCN is that GCN learn node representation by extracting the spectral information of the graph. According to Theorem 2, degree-based sampling preserves the spectrum (e.g., eigenvalues) of the input graph. Thus, the learned node representation should be similar.

Here we analyze the behaviour of GCNs with sparsified graph for simplicity. We note that our analysis can be applied to other GNNs. Let $\mathcal{G}$ be the original graph and $\mathcal{G}'$ be the graph sparsified by degree-based sampling. Specifically, let $\boldsymbol{H}^{(l)}$ and $\boldsymbol{H}'^{(l)}$ be the node embeddings learned on the original graph $\mathcal{G}$ and sparsified graph $\mathcal{G}'$ at $l$-th layer of GCNs, respectively. Let $\lambda_1$ and $\alpha$ be the largest and the smallest non-zero eigenvalue of the original $\mathcal{L}$, respectively. We have:

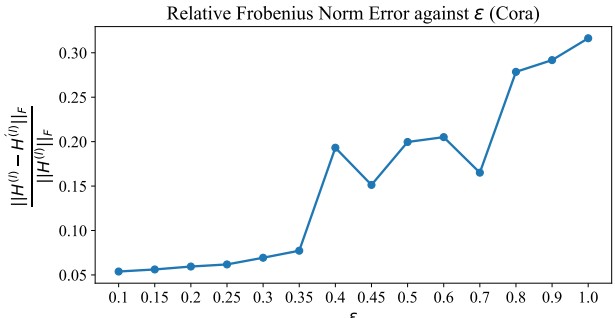

Relative Frobenius Norm Error against $\varepsilon$ (Cora)

**Theorem 3** (Proof in Appendix B).

$$\|\boldsymbol{H}^{(l+1)} - \boldsymbol{H}'^{(l+1)}\|_F < \epsilon \frac{\lambda_1}{\alpha} \|\boldsymbol{H}^{(l)} \boldsymbol{\Theta}^{(l)}\|_F. \tag{6}$$

Figure 3: The relative Frobenius norm error against $\epsilon$ on Cora dataset.

The above Theorem shows that our proposed degree-based spectral sparsification leads to good approximations of node embeddings learned on the original graphs, especially for well-connected graph.

**Experimental Analysis:** Here we experimentally validate whether our proposed degree-based spectral sparsification leads to good approximations of node embeddings learned on the original graphs. Specifically, we evaluate the same GNN on two graphs, the original and the sparsified graph. To get a sense on scale, in Figure 3 we plot the relative Frobenius norm error $\frac{\|\boldsymbol{H}^{(l+1)} - \boldsymbol{H}'^{(l+1)}\|_F}{\|\boldsymbol{H}^{(l+1)}\| \|_F}$ against the error term $\epsilon$ for Cora dataset (Sen et al., 2008), where the model is a two-layer GCN with hidden size 32. In summary, we observe for small $\epsilon$ ($\leq 0.35$), the relative Frobenius norm error is about $5\% \sim 10\%$. Then the error will grow rapidly to $30\%$ when $\epsilon = 1.0$. Thus the error is acceptable when a proper $\epsilon$ is selected. Later we experimentally show that the model accuracy drop is negligible even with a very aggressive $\epsilon$.

## 4    Related Works and Discussion

**Subgraph-based GNN training.** The key idea of this line of work is to train GNNs with sampled subgraphs to reduce the number of nodes retained in memory. Based on this idea, various sampling techniques have been proposed, including the node-wise sampling Hamilton et al. (2017); Chen et al. (2017), layer-wise sampling Huang et al. (2018); Zou et al. (2019), and subgraph sampling Chiang et al. (2019); Zeng et al. (2020). Similar to DSpar , subgraph-based methods also introduce the error during the training process. However, the key difference is that subgraph-based methods generate several subgraphs before training, and use different subgraphs at each training step. In contrast, DSpar only outputs one sparsified graph and uses it for both training and inference. Thus, methods in this category are orthogonal to DSpar . We experimentally show that DSpar can be integrated with subgraph-sampling based methods to achieve a better efficiency without loss of accuracy.

**System-level Acceleration.** This research line can be roughly divided into two categories. First, some works propose distributed GNNs training systems, which focus on minimizing the communication cost among hardware (Zheng et al., 2020b; Wan et al., 2022b;a). Second, another research line optimizes the memory access pattern of sparse operations via coalescing the memory access and fusing consecutive operations (Zhang et al., 2022; Huang et al., 2020; Rahman et al., 2021; Wang et al., 2021). We note that our work is orthogonal to system-level acceleration methods.

**Approximation.** Some works try to replace the expensive sparse operations with their cheaper approximated versions (Narayanan et al., 2022; Liu et al., 2022a;b; 2023c; Wang et al., 2022). For example, Liu et al. (2022a) replaces the original SpMM operations with their faster-but-inaccurate versions.

**Graph Sparsification.** Graph sparsification provides a "sketch" of the input graph by removing redundant edges. Traditional graph sparsification uses efficient resistance as the importance score for removing the edges Spielman & Srivastava (2011). However, as we analyzed, these methods are not practical on large graphs. Then another research line tries to learn the edge importance score. Specifically, Zheng et al. (2020a) proposes a learning-based graph sparsification method which removes potentially task-irrelevant edges from input graphs Li et al. (2020) formulates the graph sparsification problem as an optimization objective which can be solved by alternating direction method of multipliers (ADMM). Chen et al. (2021) proposes to co-simplify the input graph and GNN model by extending the iterative magnitude pruning to graph areas. However, we note that these learning-based sparsification methods have extra training process, and thus introduces significant ahead-of-training overhead. Moreover, learning-based methods are not scalable since it need to assign each edge an extra trainable mask variable, which is extremely expensive for large graphs.

### 4.1    Comparison to other sampler using node degree information

**GraphSAINT edge sampler.** Both the edge sampler in GraphSAINT and DSpar assign $p_e \propto \frac{1}{d_u} + \frac{1}{d_v}$. Here we would like to highlight three key difference between them.

- First, they generate subgraphs differently. The edge sampler in GraphSAINT is used to build **node-induced subgraph** (Zeng et al., 2020). Namely, it first selects a subset of **anchor nodes** using the edge sampler and including all the edges that connect those nodes. In other words, the induced subgraph may contain edges that are not sampled. In contrast, the graph sparsified by DSpar can be viewed as **edge-induced subgraph**. Namely, DSpar first selects a subset of edges from the original graph and includes only those nodes that are endpoints of the selected edges.

- Second, they are executed differently. One key difference between DSpar and the graph sampler (e.g., edge sampler and FastGCN sampler (Chen et al., 2018)) is that **DSpar only sparsify the graph once before training**. In contrast, the graph sampler generates different subgraphs at each training step.

- Third, they are derived differently and thus are used differently. $\frac{1}{d_u} + \frac{1}{d_v}$ in GraphSAINT edge sampler is obtained by debiasing node embeddings in the sampled subgraph (Zeng et al., 2020). We note that this bias term stems from using the node-induced subgraph. In contrast, DSpar is derived following the traditional spectral graph theory, which mainly focuses on providing one light-weight sketch of the original graph.

**The FastGCN sampler Chen et al. (2018)** shares some similarities with the DSpar method, as both techniques employ node degree information. However, they utilize this information in significantly different ways. Specifically, ***for a given batch of nodes, FastGCN samples neighbors for each in-batch node with a probability proportional to the square of the node degree.*** In contrast, DSpar uses node degree to subsample edges with a probability proportional to $\frac{1}{d_u} + \frac{1}{d_v}$ for any edge $e$ in the graph.

This difference in sampling strategies leads to distinct outcomes. In FastGCN, a neighbor with a higher degree has a greater chance of being sampled for a given node. Conversely, in the DSpar method, an edge is less likely to be sampled if the degrees of its endpoints are large. Moreover, FastGCN sampler is developed for selecting neighbors for a given node, which cannot be directly extended to the area of graph sparsification.

### 4.2 Limitations

Here we would like to briefly discuss the limitation of our work. First, unlike previous theoretical pioneer on GNN generalization (Li et al., 2022; Zhang et al., 2023), Theorem 3 is not directly related to the generalization of GNNs trained on sparsified graphs. However, we would like to note that our main theoretical result (Theorem 2) is derived without making any assumptions. Obtaining a meaningful generalization bound for GNNs without any assumptions is highly challenging. We leave it as a future work. Thus, the generalization of GNNs trained on large graphs is only experimentally verified.

## 5 Experiment

We verify the effectiveness of our proposed framework through answering the following research questions:

- **Q1**: **How effective is the proposed DSpar in terms of model accuracy and efficiency compared to the random sparsification? What is the sampling time overhead of DSpar ?**

- **Q2**: How effective is DSpar in terms of the preserved spectral information, i.e., eigenvalues?

- **Q3**: **How efficient is DSpar compared to the baseline during inference?**

- **Q4**: How sensitive is DSpar to its key hyperparameters?

### 5.1 Experimental Settings

We first introduce the applied baselines and datasets. Then we introduce the evaluation metrics for measuring the speed, and accuracy, respectively. Finally, we introduce the hyperparameter settings for DSpar . Following previous works Fey et al. (2021); Hu et al. (2020); Duan et al. (2022), **we focus on the transductive node classification**, which is also the most common task in large-scale graph benchmarks.

#### 5.1.1 Datasets and Baselines

To evaluate DSpar , we adopt four common large scale graph benchmark datasets from different domains, namely, **Reddit Hamilton et al. (2017), Yelp Zeng et al. (2020),** *ogbn-arxiv* **Hu et al. (2020),** *ogbn-proteins* **Hu et al. (2020) and** *ogbn-products* **Hu et al. (2020).** We evaluate DSpar under both the mini-batch training and full-batch training settings. For the mini-batch training setting, we integrate DSpar with the state-of-the-art subgraph sampling methods, i.e., GraphSAINT Zeng et al. (2020). **When integrating DSpar with subgraph sampling methods, we first sparsify the input graph by Algorithm 1 and then sample subgraphs from the sparsified graph.** For the full-batch training setting, we integrate DSpar with three popular models: two commonly used shallow models, namely, GCN Kipf & Welling (2017) and GraphSAGE Hamilton et al. (2017), and one deep model GCNII Chen et al. (2020). **To avoid creating confusion, GCN, GraphSAGE, and GCNII are all trained with the whole graph at each step**. For a fair comparison, we use the mean aggregator for GraphSAGE and GraphSAINT throughout the paper. Details about the hyperparameters are in Appendix C.

We compare DSpar with random sparsification ("random" in Table 1), where each edge has equal probability to be removed. We note that we only compare to the random sparsification mainly because (1) learning-based

Table 1: Comparison of test accuracy (↑) and training throughput (↑) on five datasets. Gray cells indicate the accuracy drop is negligible (≈ 0.3%) or the result is better compared to the baseline. The hardware here is a single NVIDIA A40 (48GB). All reported results are averaged over ten random trials.

| | | # nodes | 230K | # edges | 11.6M | | | | | | | | |
|---|---|---|---|---|---|---|---|---|---|---|---|---|---|

| | | 230K | | 717K | | 169K | | 132K | | 2.4M | | | |
| | | 11.6M | | 7.9M | | 1.2M | | 39.5M | | 61.9M | | | |
| Model | Methods | Reddit | | Yelp | | *ogbn-arxiv* | | *ogbn-proteins* | | *ogbn-products* | | Avg. | |
| | | Acc. | Throughput (epoch/s) | F1-micro | Throughput (epoch/s) | Acc. | Throughput (epoch/s) | ROC-AUC. | Throughput (epoch/s) | Acc. | Throughput (epoch/s) | Δ Acc. | Speedup |
| Graph-SAINT | Baseline | 96.02±0.08 | 3.39 | 63.78±0.12 | 0.72 | 71.49±0.20 | 12.94 | 75.54±0.40 | 0.30 | 79.03±0.23 | 0.19 | 0.0 | 1.0× |
| | +random | 94.46±0.11 | 3.31 | 63.68±0.06 | 0.80 | 71.02±0.23 | 13.15 | 75.99±0.06 | 0.61 | 78.40±0.37 | 0.33 | ↓0.45 | 1.4× |
| | +DSpar | 96.11±0.07 | 3.76 (1.1 ×) | 63.91±0.14 | 0.79 (1.1×) | 71.40±0.09 | 13.81 (1.1×) | 75.66±0.20 | 0.61 (2×) | 78.97±0.35 | 0.32 (1.7×) | ↑0.04 | 1.4× |
| GCN | Baseline | 95.39±0.04 | 7.69 | 40.22±0.47 | 4.07 | 71.87±0.16 | 36.14 | 71.99±0.66 | 3.92 | 75.74±0.11 | 0.50 | 0.0 | 1.0× |
| | +random | 94.22±0.03 | 17.01 | 40.32±0.58 | 5.21 | 70.89±0.13 | 37.6 | 72.14±0.67 | 21.16 | 73.33±0.11 | 1.85 | ↓0.9 | 2.8× |
| | +DSpar | 95.33±0.03 | 16.71 (2.2×) | 41.01±0.18 | 5.42 (1.3×) | 71.72±0.25 | 39.01 (1.1×) | 72.65±0.52 | 22.93 (5.8×) | 75.69±0.07 | 1.88 (3.8×) | ↑0.26 | 2.8× |
| Graph-SAGE (full batch) | Baseline | 96.44±0.04 | 4.33 | 62.05±0.14 | 3.66 | 71.85±0.24 | 32.28 | 76.09±0.77 | 3.87 | 78.78±0.19 | 0.65 | 0.0 | 1.0× |
| | random | 94.97±0.04 | 10.29 | 62.00±0.21 | 4.79 | 71.26±0.32 | 33.53 | 75.88±0.21 | 21.92 | 74.03±0.22 | 1.67 | ↓1.39 | 2.6× |
| | +DSpar | 96.45±0.04 | 9.97 (2.3×) | 61.86±0.15 | 4.81 (1.3×) | 71.94±0.24 | 34.34 (1.1×) | 76.71±0.09 | 23.21 (5.9×) | 78.84±0.12 | 1.67 (2.6×) | ↑0.12 | 2.6× |
| GCNII | Baseline | 96.71±0.07 | 2.20 | 64.02±0.13 | 0.84 | 72.85±0.27 | 2.13 | 73.79±1.32 | 1.75 | — | — | 0.0 | 1.0× |
| | +random | 95.66±0.03 | 4.06 | 63.59±0.11 | 0.97 | 72.29±0.35 | 2.18 | 73.85±0.51 | 9.74 | — | — | ↓0.50 | 2.5× |
| | +DSpar | 96.65±0.06 | 3.99 (1.8×) | 63.98±0.09 | 0.99 (1.2×) | 72.58±0.51 | 2.22 (1.1×) | 74.09±0.61 | 10.33 (5.9×) | — | — | ↓0.02 | 2.5× |

sparsification introduces significant ahead-of-training overhead because it requires extra learning process to identify unimportant edges. Moreover, learning-based methods are not scalable since it need to assign each edge an extra trainable mask variable, which is extremely expensive for large graphs Chen et al. (2021). (2) estimating effective resistance for all edges in large graphs is not practical.

### 5.1.2 Evaluation metrics

We comprehensively investigate the practical usefulness of our proposed method by evaluating the trade-off between the speed and accuracy. Specifically,

**Speed**: Based on FLOPs, `SpMM` is theoretically much faster than `Matmul`. However, this is not true in practice due to the random memory access pattern of `SpMM`, which cannot be efficiently accelerated on CPUs and GPUs Han et al. (2016). To evaluate the practical usage of our method, we measure the actual running speed on the off-the-shell hardwares. For **Training speed**, we measure the hardware throughput on GPUs (epoch/s). For **Inference speed**, we measure the latency on GPUs (ms). For **Ahead-of-training overhead**, we measure the wall clock time (s).

**Accuracy**: Following Hu et al. (2020); Zeng et al. (2020), we use the test accuracy for evaluating performance on Reddit, *ogbn-arxiv*, and *ogbn-products* datasets. The F1-micro is used for Yelp dataset. The ROC-AUC is used for *ogbn-proteins* dataset.

### 5.1.3 Hyperparameter Settings

For graph sparsification, the hyperparameter is the number of trials $Q$ in Algorithm 1. According to Theorem 2, we set $Q = \frac{|\mathcal{V}| \log |\mathcal{V}|}{\epsilon^2}$, where $\epsilon$ controls the sparsity and the approximation errors. We alter the value of $\epsilon$ to see how it affects the accuracy drop. Specifically, we vary $\epsilon$ from 0.3 to 1.5.

### 5.2 Accuracy versus Training Efficiency (Table 1)

### 5.2.1 The training efficiency

To answer **Q1**, we summarize the training throughput and the accuracy of different methods in Table 1. We also report the model performance under random sparsification, i,e., "random" in Table 1. **For a fair comparison, we control the number of removed edges roughly the same for both DSpar and random sparsification. In this way, they should have the same acceleration effect.** We show the sparsification effect of DSpar in Table 2. The ahead-of-training overhead of DSpar is given in Table 3. We also show how is the optimization process affected by DSpar in Figure 4. We observe:

- ❶ Regardless of subgraph training (i.e., GraphSAINT) or full graph training (i.e., GCN, GraphSAGE, and GCNII), **the accuracy drop of applying DSpar over baselines is negligible (≈ 0.3%) across**

Table 2: The effect of DSpar (Algorithm 1), which removes $\approx 25 \sim 95\%$ edges for different datasets, according to graph statistics. When applying DSpar over subgraph sampling methods, the sparsity may need to be decreased to guarantee the accuracy since subgraph sampling methods also introduce extra error.

| Dataset | Graph Sparsification Rate $(= 1 - \frac{|\mathcal{E}'|}{|\mathcal{E}|})$ | | | | |
| | Reddit | Yelp | *ogbn-arxiv* | *ogbn-proteins* | *ogbn-products* |
| --- | --- | --- | --- | --- | --- |
| Graph-SAINT | 82.1% | 73.5% | 26.7% | 93.1% | 31.9% |
| GCN | 82.1% | 73.5% | 26.7% | 95.0% | 79.4% |
| Graph-SAGE | 82.1% | 73.5% | 26.7% | 95.0% | 79.4% |
| GCNII | 82.1% | 73.5% | 26.7% | 95.0% | — |

Table 3: Ahead-of-training overhead (sampling time) of DSpar

| Dataset | Graph Sparsification Rate $(= 1 - \frac{|\mathcal{E}'|}{|\mathcal{E}|})$ (%) | Sampling time (s) |
| --- | --- | --- |
| Reddit | 82.1 | 2.05 |
| Yelp | 73.5 | 1.62 |
| *ogbn-arxiv* | 26.7 | 1.4 |
| *ogbn-proteins* | 95.0 | 1.5 |
| *ogbn-products* | 79.4 | 11.0 |

**different models and datasets.** This can be explained by Theorem 3 that GNNs still learn expressive node representation on graphs sparsified by DSpar . As we mentioned at the beginning of Section 5.2, we compare the model accuracy drop of DSpar and random sparsification under the same speedup. Interestingly, for Yelp and *ogbn-proteins*, random sparsification performs almost the same compared to DSpar . However, the accuracy drop of random sparsification is much larger on other datasets, especially on *ogbn-products*. We hypothesize this might because for Yelp and *ogbn-proteins*, the node features are more important to the model accuracy compared to the edge connectivity.

- ❷ **DSpar can significantly reduce the training time of GNNs with even better accuracy.** As shown in Table 1, we summarize the training throughput (epoch/s) and speedup over baseline on a single NVIDIA A40 GPU. Notably, DSpar achieves non-trivial trade-off between accuracy and efficiency. Namely, in average DSpar brings $\approx 2.5\times$ speedup for full-graph training and $1.4\times$ speedup for mini-batch training, with almost no accuracy drop or even better accuracy. The better accuracy can be explained by the fact that removing edges may impose regularization effect during training Rong et al. (2019).

- ❸ **DSpar removes $\approx 25\% \sim 95\%$ edges across different datasets, with negligible ahead-of-time overhead.** As shown in Table 2, we summarize the sparsification effect of DSpar . According to Theorem 2 and Theorem 3, $Q$ in Algorithm 1 should be scale with the graph connectivity, measure by the ratio of largest eigenvalue $\lambda_1$ to the smallest non-zero eigenvalue $\alpha$ of the graph Laplacian matrix. In practice, we absorb the term of $\frac{\lambda}{\alpha}$ into $\epsilon$. Specifically, we control the sparsify through directly tuning $\epsilon$ with $Q = \frac{|\mathcal{V}| \log |\mathcal{V}|}{\epsilon^2}$. When applying DSpar over subgraph sampling methods, the sparsity may need to be decreased to guarantee the accuracy since subgraph sampling methods also introduce extra error. For the ahead-of-training overhead, i.e., the running time of Algorithm 1, is reported in Table 3. **We note that we provide an optimized sampler for sampling edges in large graphs, details are elaborated in Appendix A.** We summarize that the overhead is negligible compared to the acceleration effect of graph sparsification.

- ❹ **DSpar almost has the same convergence behaviour compared to the baseline.** We investigate the convergence speed of DSpar , where the convergence speed is measured by the gap in training loss between consecutive epochs. Figure 4 summarizes the training curves of GNNs trained with different

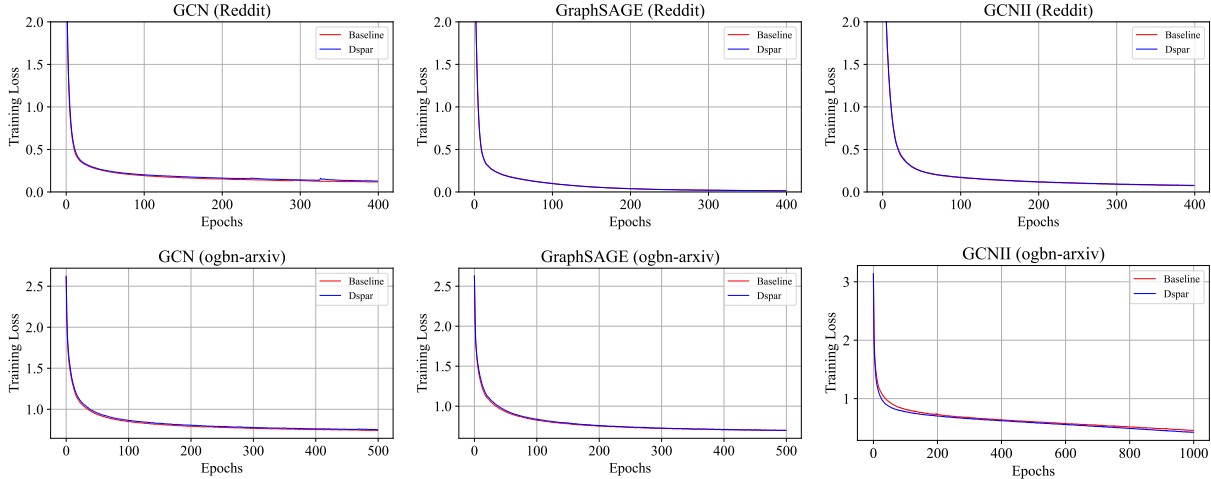

Figure 4: Training loss on Reddit and *ogbn-arxiv* dataset with different methods.

methods on Reddit and *ogbn-arxiv* dataset. We observe that DSpar almost has the same convergence behaviour compared to the baseline. This is consistent with our theoretical analysis that DSpar leads to good approximations of node embeddings learned on the original graphs.

### 5.2.2 Can DSpar preserves the graph spectral information, i.e., eigenvalues?

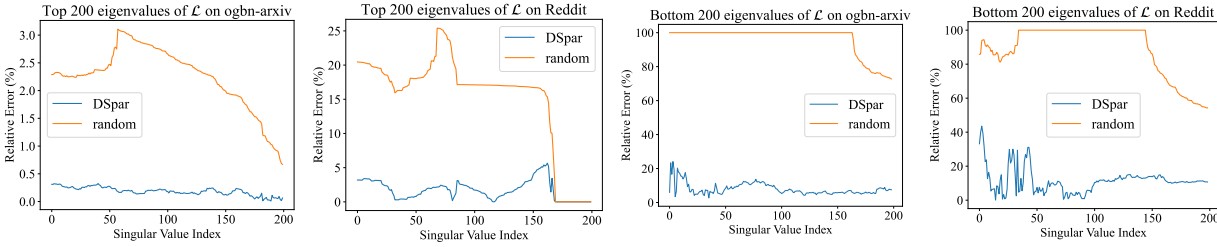

Figure 5: The relative error of the top and bottom eigenvalue of $\mathcal{L}$, i.e., $\frac{\lambda_i - \lambda_i'}{\lambda_i}$, sparsified by different methods.

As we analyzed, preserving graph spectral information is crucial for learning meaningful representations on graphs. To answer **Q2**, we calculate the relative error of the eigenvalue of the graph Laplacian matrix $\mathcal{L}$ on *ogbn-arxiv* and Reddit dataset, respectively. **We note that it is almost impossible to calculate all of the eigenvalues of a large graph. As we mentioned, small (large) eigenvalues indicate the global clustering (local smoothness) structure of the graphs. Thus in Figure 5, we instead calculate the top-200 and bottom-200 eigenvalues of $\mathcal{L}$ corresponding to the original graph and sparsified graph, respectively.** We observe:

- ❺ **DSpar preserves most of the key eigenvalues.** Specifically, we observe that for both *ogbn-arxiv* and Reddit dataset, DSpar significantly outperforms the random sparsification in terms of the relative error of eigenvalues. For the top 200 eigenvalues, which indicate the global clustering structure, DSpar provides an accurate approximation. That means the cluster structure is well-preserved by DSpar . However for the bottom 200 eigenvalues, which indicate the local smoothness, DSpar has $\approx 5 \sim 40\%$ relative errors for these bottom eigenvalues. In contrast, random sparsification almost has 100% relative error, i.e., failing to preserve this part of information. Our analysis also partially explains why GNNs learned on graph with random sparsification have much worse performance on these two datasets.

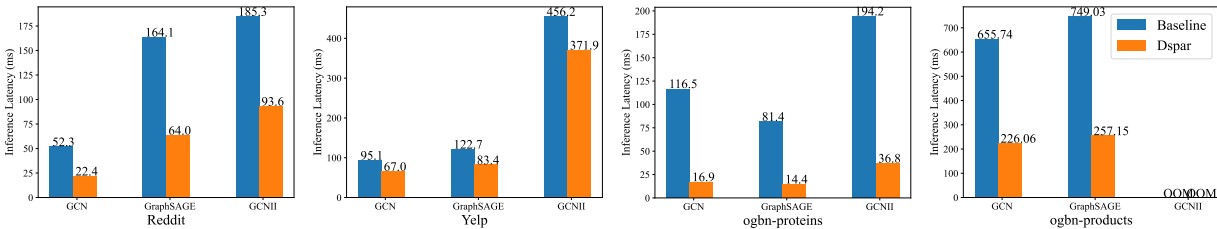

Figure 6: Inference latency comparison on a single NVIDIA A40 (48GB) GPU (lower is better). "OOM" means out-of-memory. Compared to the baseline, DSpar reduce $\approx 30\% \sim 90\%$ inference latency.

### 5.2.3  The Inference efficiency

For GraphSAINT, the subgraph sampling is only applied to the training process. Thus, they do not affect the inference latency of GNNs. To answer **Q3**, Figure 6 compares the inference latency of different models on the original graph and the graph sparsified by DSpar . We observe that

- ❺ **According to Figure 6, DSpar reduces** $\approx 30\% \sim 90\%$ **inference latency**. Notably, on Yelp dataset, DSpar reduces only 30% inference latency, although it removes more than 70% edges (Table 2). This is mainly because for Yelp dataset, graph-based operations account for a relatively small percentage of the total time compared to other datasets. For other datasets, we observe that DSpar significantly reduces up to $\approx 90\%$ inference latency.

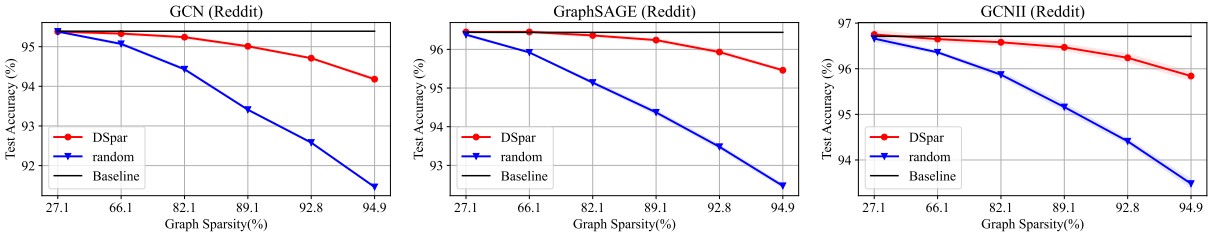

Figure 7: Accuracy versus the graph sparsity on Reddit dataset. Here the graph sparsity equals the percentage of removed edges. All results are averaged over ten random trials.

### 5.3  Hyperparameter sensitivity analysis

As we analyzed, DSpar has only one hyperparameter, namely, the number of trial $Q$ for controlling the graph sparsity. In this subsection, to answer **Q4**, we explore the sensitivity of hyperparameters $Q$ for DSpar . As we mentioned, we set $Q = \frac{|\mathcal{V}| \log |\mathcal{V}|}{\epsilon^2}$ according to Theorem 3, where $\epsilon$ controls the approximation error. We alter the value of $\epsilon$ from 0.3 to 1.5 to check the relationship between the graph sparsity and the accuracy drop. As shown in Figure 7, the accuracy drop becomes larger when edges are being removed. This is consistent with the theoretical analysis in Section 3.3 that we have larger approximation errors when the graph sparsity is high. In addition to studying dense graphs like Reddit, we also performed a sparsification experiment on a large sparse graph, specifically ogbn-products, us-

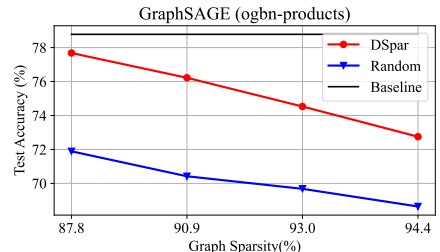

Figure 8: Accuracy versus the graph sparsity on ogbn-products.

ing GraphSAGE. We observe that in this case, the accuracy drop of random sparsification and DSpar is more significant. This is mainly because ogbn-products is much sparser than Reddit, making it more sensitive to edge removal. We also observe that DSpar significantly outperforms the random baseline, which is consistent

with the theoretical analysis. Furthermore, when we remove roughly 95% of edges in Reddit dataset, the accuracy drop is roughly $0.8 \sim 1.2\%$ for different models, which is still acceptable when considering a 95% sparsity. **In practice, we suggest selecting $Q = \frac{|\mathcal{V}| \log |\mathcal{V}|}{\epsilon^2}$ according to the accuracy drop and the efficiency constraint by adjusting $\epsilon$.**

## 6 Conclusions and Future work

We propose DSpar , a simple-yet-effective framework for training GNNs with compressed tensors, which can be plugged into most of the existing solutions to save memory. We demonstrate the potential of DSpar for the practical usage by systematically evaluating the trade-off among the memory-saving, time overhead, and accuracy drop. Future work includes (1) evaluating DSpar under the distributed training setting; (2) exploring other types of graph sparsification methods.

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

# A    Implementations

We use CPUs for sparsifying graph using Algorithm 1, which because the whole graph may exceed the GPU memory. We emphasize that Algorithm 1 sparsifies the graph by sampling with replacement (line 3). Thus, the sampling process can be easily paralleled. In practice,to avoid the numerical precision problem for graphs with more than $2^{24}$ edges [1], we use `torch.double` as the precision for the probability tensor $\{p_e\}_{e\in\mathcal{E}}$, which is not supported for most of the built-in sampling methods in Pytorch. Thus, we implement Algorithm 1 in C++ and parallel it base on OpenMP. We build it as an extension for Pytorch, which can be called using Pytorch API.

# B    Theory

## B.1    Proof of Theorem 2

**Theorem 2** (Proof in Appendix B). *Given an input graph* $\mathcal{G} = (\mathcal{V}, \mathcal{E})$, *let* $\boldsymbol{A}$ *be the associated adjacency matrix and* $\alpha$ *be the smallest non-zero eigenvalue of* $\mathcal{L}$. *Given an error parameter* $\epsilon$, *If we set* $Q = \mathcal{O}(\frac{|\mathcal{V}|\log|\mathcal{V}|}{\epsilon^2})$ *and for each edge* $e = (u,v)$, *we set* $p_e \propto \frac{1}{d_u} + \frac{1}{d_v}$, *Algorithm 1 produces a sparsified graph* $\mathcal{G}' = (\mathcal{V}', \mathcal{E}')$ *with* $\boldsymbol{A}'$, *for any vector* $\boldsymbol{x} \in \mathbb{R}^{|\mathcal{V}|}$, *we have*

$$(1 - \frac{\epsilon}{\alpha}) \sum_{(u,v)\in\mathcal{E}} (x_u - x_v)^2 A_{u,v} \leq \sum_{(u,v)\in\mathcal{E}'} (x_u - x_v)^2 A'_{u,v} \leq (1 + \frac{\epsilon}{\alpha}) \sum_{(u,v)\in\mathcal{E}} (x_u - x_v)^2 A_{u,v}. \tag{4}$$

*Proof.* Below we first introduce the necessary concepts and tools for deriving Theorem 2. Let $\boldsymbol{B} \in \mathbb{R}^{|\mathcal{E}|\times|\mathcal{V}|}$ be the signed edge-vertex incidence matrix, given by

$$B_{(e,v)} = \begin{cases} 1 & \text{if } v \text{ is edge } e\text{'s head,} \\ -1 & \text{if } v \text{ is edge } e\text{'s tail,} \\ 0 & \text{otherwise.} \end{cases} \tag{7}$$

Let $\boldsymbol{W} \in \mathbb{R}^{|\mathcal{E}|\times|\mathcal{E}|}$ be a diagnoral matrix such that $\boldsymbol{W}_{e,e} = \boldsymbol{A}_e$. Then we define the projection matrix $\Pi$ as $\Pi = \boldsymbol{W}^{\frac{1}{2}}\boldsymbol{B}\mathcal{L}^{+}\boldsymbol{B}^{\top}\boldsymbol{W}^{\frac{1}{2}}$. We note that $\Pi$ have the following properties Spielman & Srivastava (2011):

- $\Pi$ is a diagonal projection matrix,

- $\|\Pi_{\cdot,e}\|_2^2 = \Pi_{e,e}$,

- $\Pi\Pi = \Pi$,

- $\Pi_{e,e} = A_e R_e$.

Let $\boldsymbol{S} \in \mathbb{R}^{\mathcal{E}}$ be the diagonal random matrix representing the sampling process in Algorithm 1 such that

$$\boldsymbol{S}_{e,e} = \frac{(\#\text{of times e is sampled})}{Qp_e} \tag{8}$$

Then we have the following Lemma:

**Lemma 1** (Lemma 4 in Spielman & Srivastava (2011)). *Let* $\boldsymbol{S}$ *be the diagonal random matrix representing the sampling process such that*

$$\|\Pi\boldsymbol{S}\Pi - \Pi\Pi\| \leq \epsilon,$$

*Then*

$$\forall \boldsymbol{x} \in \mathbb{R}^{|\mathcal{V}|}, (1 - \epsilon)\boldsymbol{x}^{\top}\mathcal{L}\boldsymbol{x} \leq \boldsymbol{x}^{\top}\mathcal{L}'\boldsymbol{x} \leq (1 + \epsilon)\boldsymbol{x}^{\top}\mathcal{L}\boldsymbol{x}, \tag{9}$$

*where* $\mathcal{L} = \boldsymbol{B}^{\top}\boldsymbol{W}\boldsymbol{B}$ *is the normalized Laplacian matrix and* $\mathcal{L}' = \boldsymbol{B}^{\top}\boldsymbol{W}^{\frac{1}{2}}\boldsymbol{S}\boldsymbol{W}^{\frac{1}{2}}\boldsymbol{B}$ *is the sparsified Laplacian matrix .*

---

[1]https://www.mathworks.com/help/matlab/ref/flintmax.html

Sampling $Q$ edges from $\mathcal{G}$ corresponds to sampling $Q$ columns from $\Pi$, so we can write

$$\Pi \boldsymbol{S} \Pi = \sum_e \boldsymbol{S}_{e,e} \Pi_{:,e} \Pi_{:,e}^\top \tag{10}$$

$$= \sum_e \frac{(\#\text{of times e is sampled})}{Q p_e} \Pi_{:,e} \Pi_{:,e}^\top \tag{11}$$

$$= \frac{1}{Q} \sum_e (\#\text{of times e is sampled}) \frac{\Pi_{:,e}}{\sqrt{p_e}} \frac{\Pi_{:,e}^\top}{\sqrt{p_e}} \tag{12}$$

$$= \frac{1}{Q} \sum_{i=1}^Q \boldsymbol{y}_i \boldsymbol{y}_i^\top, \tag{13}$$

where $\boldsymbol{y}_1, \cdots, \boldsymbol{y}_Q$ are drawn independently with replacement from the distribution

$$\boldsymbol{y} = \frac{\Pi_{:,e}}{\sqrt{p_e}} \quad \text{with probability } p_e. \tag{14}$$

Recall that according to Theorem 1, we know that for any edge $e = (u,v)$, we have $\frac{1}{2}(\frac{1}{d_u} + \frac{1}{d_v}) \le R_e \le \frac{1}{\alpha}(\frac{1}{d_u} + \frac{1}{d_v})$. Let $p_e$ and $p'_e$ be the sampling probability for edge $e$ in proportional to $R_e$ and $\frac{1}{d_u} + \frac{1}{d_v}$, respectively. Let $\boldsymbol{S}$ and $\boldsymbol{S}'$ be the random matrix with $p_e$ and $p'_e$, respectively. Let $\boldsymbol{y}$ and $\boldsymbol{y}'$ be the distribution defined above associated with $p_e$ and $p'_e$, respectively. Then

$$p'_e = \frac{\frac{1}{d_u} + \frac{1}{d_v}}{\sum_{e=(u,v)\in\mathcal{E}} \frac{1}{d_u} + \frac{1}{d_v}} \ge \frac{\alpha R_e}{2 \sum_{e=(u,v)\in\mathcal{E}} R_e} = \frac{\alpha}{2} p_e. \tag{15}$$

For the norm of $\boldsymbol{y}$, we have:

$$\|\boldsymbol{y}\|_2^2 = \frac{1}{p_e} \|\Pi_{:,e}\|_2^2 = \frac{1}{p_e} \Pi_{e,e}^2 = \frac{\sum_e R_e}{R_e} R_e = \sum_e R_e = |\mathcal{V}| - 1.$$

We note that we have $\sum_e R_e = |\mathcal{V}| - 1$ from the definition of effective resistance Spielman & Srivastava (2011).

Then we have

$$\|\boldsymbol{y}'\|_2^2 = \frac{\|\Pi_{:,e}\|_2^2}{p'_e} \le \frac{2}{\alpha} \frac{\|\Pi_{:,e}\|_2^2}{p_e} = \frac{2}{\alpha} \|\boldsymbol{y}\|_2^2 \le \frac{2}{\alpha}(|\mathcal{V}| - 1) \tag{16}$$

$$\mathbb{E}[\Pi \boldsymbol{S}' \Pi - \Pi\Pi] = \mathbb{E}[\frac{1}{Q} \sum_{i=1}^Q \boldsymbol{y}'_i \boldsymbol{y}'_i^\top - \mathbb{E}\boldsymbol{y}' \boldsymbol{y}'^\top] \tag{17}$$

**Lemma 2** (Lemma 5 in Spielman & Srivastava (2011)). *Let $\boldsymbol{p}$ be the probability distribution over $\Omega \in \mathbb{R}^{|\mathcal{V}|}$ such that $\|\boldsymbol{y}\|_2 \le M$ and $\|\mathbb{E}\boldsymbol{y}\boldsymbol{y}^\top\| \le 1$, we have*

$$\mathbb{E}\|\frac{1}{Q} \sum_{i=1}^Q \boldsymbol{y}_i \boldsymbol{y}_i^\top - \mathbb{E}\boldsymbol{y}\boldsymbol{y}^\top\|_2 \le \min(CM\sqrt{\frac{\log Q}{Q}}, 1), \tag{18}$$

*where $C$ is an absolute constant.*

By Taking $Q = 9C^2 |\mathcal{V}| \log |\mathcal{V}| / \epsilon^2$, we have

$$\mathbb{E}[\Pi \boldsymbol{S}'\Pi - \Pi\Pi] \le \frac{\epsilon}{\alpha}, \tag{19}$$

By Lemma 1, this completes the proof of the theorem.

$\square$

### B.2 Derive Equation (5)

By Courant-Fischer Theorem, we know that the $i$-th eigenvalue of $\mathcal{L}$ satisfies

$$\lambda_i = \min_{\boldsymbol{K}:dim(\boldsymbol{K})=i} \max_{\boldsymbol{x}\in\boldsymbol{K}} \frac{\boldsymbol{x}^\top \mathcal{L} \boldsymbol{x}}{\boldsymbol{x}^\top \boldsymbol{x}}, \tag{20}$$

where $dim(\boldsymbol{K})$ is the number of linearly independent vectors that form a basis for $\boldsymbol{K}$.

Similarly, for $\mathcal{L}'$, the $i$-th eigenvalue $\lambda_i'$ can be expressed as:

$$\lambda_i' = \min_{\boldsymbol{K}:dim(\boldsymbol{K})=i} \max_{\boldsymbol{x}\in\boldsymbol{K}} \frac{\boldsymbol{x}^\top \mathcal{L}' \boldsymbol{x}}{\boldsymbol{x}^\top \boldsymbol{x}}. \tag{21}$$

Now, by Theorem 2, we know that for any $\boldsymbol{x}$, we have

$$(1 - \frac{\epsilon}{\alpha})\boldsymbol{x}^\top \mathcal{L} \boldsymbol{x} \le \boldsymbol{x}^\top \mathcal{L}' \boldsymbol{x} \le (1 + \frac{\epsilon}{\alpha})\boldsymbol{x}^\top \mathcal{L} \boldsymbol{x}, \tag{22}$$

Let $\boldsymbol{K}_i$ be any subspace with dimension $i$. Then, for any vector $\boldsymbol{x} \in \boldsymbol{K}_i$, we have:

$$(1 - \frac{\epsilon}{\alpha})\frac{\boldsymbol{x}^\top \mathcal{L} \boldsymbol{x}}{\boldsymbol{x}^\top \boldsymbol{x}} \le \frac{\boldsymbol{x}^\top \mathcal{L}' \boldsymbol{x}}{\boldsymbol{x}^\top \boldsymbol{x}} \le (1 + \frac{\epsilon}{\alpha})\frac{\boldsymbol{x}^\top \mathcal{L} \boldsymbol{x}}{\boldsymbol{x}^\top \boldsymbol{x}}. \tag{23}$$

Suppose $\frac{\boldsymbol{x}^\top \mathcal{L}' \boldsymbol{x}}{\boldsymbol{x}^\top \boldsymbol{x}}$ achieves maximum at $\boldsymbol{x}'$, now we have

$$\max_{\boldsymbol{x}\in\boldsymbol{K}_i} \frac{\boldsymbol{x}^\top \mathcal{L}' \boldsymbol{x}}{\boldsymbol{x}^\top \boldsymbol{x}} = \frac{\boldsymbol{x}'^\top \mathcal{L}' \boldsymbol{x}'}{\boldsymbol{x}'^\top \boldsymbol{x}'} \tag{24}$$

$$\le (1 + \frac{\epsilon}{\alpha})\frac{\boldsymbol{x}'^\top \mathcal{L} \boldsymbol{x}'}{\boldsymbol{x}'^\top \boldsymbol{x}'} \tag{25}$$

$$\le (1 + \frac{\epsilon}{\alpha}) \max_{\boldsymbol{x}\in\boldsymbol{K}_i} \frac{\boldsymbol{x}^\top \mathcal{L} \boldsymbol{x}}{\boldsymbol{x}^\top \boldsymbol{x}} \tag{26}$$

By applying the similar technique on the left side of the inequality, we have $(1 - \frac{\epsilon}{\alpha}) \max_{\boldsymbol{x}\in\boldsymbol{K}_i} \frac{\boldsymbol{x}^\top \mathcal{L} \boldsymbol{x}}{\boldsymbol{x}^\top \boldsymbol{x}} \le \max_{\boldsymbol{x}\in\boldsymbol{K}_i} \frac{\boldsymbol{x}^\top \mathcal{L}' \boldsymbol{x}}{\boldsymbol{x}^\top \boldsymbol{x}}$. Combine the left side and right side inequality together, we have:

$$(1 - \frac{\epsilon}{\alpha}) \max_{\boldsymbol{x}\in\boldsymbol{K}_i} \frac{\boldsymbol{x}^\top \mathcal{L} \boldsymbol{x}}{\boldsymbol{x}^\top \boldsymbol{x}} \le \max_{\boldsymbol{x}\in\boldsymbol{K}_i} \frac{\boldsymbol{x}^\top \mathcal{L}' \boldsymbol{x}}{\boldsymbol{x}^\top \boldsymbol{x}} \le (1 + \frac{\epsilon}{\alpha}) \max_{\boldsymbol{x}\in\boldsymbol{K}_i} \frac{\boldsymbol{x}^\top \mathcal{L} \boldsymbol{x}}{\boldsymbol{x}^\top \boldsymbol{x}}. \tag{27}$$

Please note that the above inequality holds for any subspace $\boldsymbol{K}_i$ with dimension $i$. Let $\hat{\boldsymbol{K}}_i$ be the subspace that achieves the minimum in the Courant-Fischer theorem for eigenvalue $\lambda_i$ in Equation (20). We have

$$\max_{\boldsymbol{x}\in\hat{\boldsymbol{K}}_i} \frac{\boldsymbol{x}^\top \mathcal{L}' \boldsymbol{x}}{\boldsymbol{x}^\top \boldsymbol{x}} \le (1 + \frac{\epsilon}{\alpha}) \max_{\boldsymbol{x}\in\hat{\boldsymbol{K}}_i} \frac{\boldsymbol{x}^\top \mathcal{L} \boldsymbol{x}}{\boldsymbol{x}^\top \boldsymbol{x}} = (1 + \frac{\epsilon}{\alpha})\lambda_i, \tag{28}$$

Now by definition of the minimum, we know that

$$\max_{\boldsymbol{x} \in \hat{\boldsymbol{K}}_i} \frac{\boldsymbol{x}^\top \mathcal{L}' \boldsymbol{x}}{\boldsymbol{x}^\top \boldsymbol{x}} \geq \min_{\boldsymbol{K}_i : dim(\boldsymbol{K}_i) = i} \max_{\boldsymbol{x} \in \boldsymbol{K}_i} \frac{\boldsymbol{x}^\top \mathcal{L}' \boldsymbol{x}}{\boldsymbol{x}^\top \boldsymbol{x}} = \lambda_i' \tag{29}$$

By connecting the above two inequality together, we have

$$\lambda_i' \leq (1 + \frac{\epsilon}{\alpha})\lambda_i. \tag{30}$$

Similarly, by applying the same technique on the left side, we can obtain

$$(1 - \frac{\epsilon}{\alpha})\lambda_i \leq \lambda_i' \tag{31}$$

By connecting the above inequality together, we obtain

$$(1 - \frac{\epsilon}{\alpha})\lambda_i \leq \lambda_i' \leq (1 + \frac{\epsilon}{\alpha})\lambda_i. \tag{32}$$

### B.3 Proof of Theorem 3

Theorem 3 is the direct extension of Theorem 2 to GNN area. In this proof, we aim to establish a bound on the difference between the activations of two GCNs with different Laplacian matrices. We compute the Frobenius norm of the difference between these activations and derive an upper bound by applying the triangle inequality and the properties of the matrix 2-norm.

**Theorem 3** (Proof in Appendix B).

$$\|\boldsymbol{H}^{(l+1)} - \boldsymbol{H}'^{(l+1)}\|_F < \epsilon \frac{\lambda_1}{\alpha} \|\boldsymbol{H}^{(l)} \boldsymbol{\Theta}^{(l)}\|_F. \tag{6}$$

*Proof.* First, by the defination of GCN, we have

$$\boldsymbol{H}^{(l+1)} = \text{ReLU}\big((2\boldsymbol{I} - \mathcal{L})\boldsymbol{H}^{(l)} \boldsymbol{\Theta}^{(l)}\big), \tag{33}$$
$$\boldsymbol{H}'^{(l+1)} = \text{ReLU}\big((2\boldsymbol{I} - \mathcal{L}')\boldsymbol{H}^{(l)} \boldsymbol{\Theta}^{(l)}\big). \tag{34}$$

Then we have

$$\begin{aligned}
\|\boldsymbol{H}^{(l+1)} - \boldsymbol{H}'^{(l+1)}\|_F &\leq \|(\mathcal{L} - \mathcal{L}')\boldsymbol{H}^{(l)} \boldsymbol{\Theta}^{(l)}\|_F \\
&\leq \|(\mathcal{L} - \mathcal{L}')\|_2 \|\boldsymbol{H}^{(l)} \boldsymbol{\Theta}^{(l)}\|_F
\end{aligned} \tag{35}$$

The above inequality directly from the fact that for any two matrix $\boldsymbol{X}$ and $\boldsymbol{Y}$, we have $\|\boldsymbol{X}\boldsymbol{Y}\|_F \leq \|\boldsymbol{X}\|_2 \|\boldsymbol{Y}\|_F \leq \|\boldsymbol{X}\|_F \|\boldsymbol{Y}\|_F$.

By definition of 2-norm, we have

$$\|(\mathcal{L} - \mathcal{L}')\|_2 = \sup_{\boldsymbol{x} : \|\boldsymbol{x}\| = 1} \boldsymbol{x}^\top (\mathcal{L} - \mathcal{L}')\boldsymbol{x} \tag{36}$$

Suppose the above supremum is achieved at $\boldsymbol{x}_0$, then

$$\|(\mathcal{L} - \mathcal{L}')\|_2 = \boldsymbol{x}_0^\top \mathcal{L} \boldsymbol{x}_0 - \boldsymbol{x}_0^\top \mathcal{L}' \boldsymbol{x}_0$$

$$\leq \boldsymbol{x}_0^\top \mathcal{L} \boldsymbol{x}_0 - (1 - \frac{\epsilon}{\alpha}) \boldsymbol{x}_0^\top \mathcal{L} \boldsymbol{x}_0 \qquad \text{(From Theorem 2)} \tag{37}$$

$$= \frac{\epsilon}{\alpha} \boldsymbol{x}_0^\top \mathcal{L} \boldsymbol{x}_0$$

$$\leq \sup_{\boldsymbol{x}:\|\boldsymbol{x}\|=1} \frac{\epsilon}{\alpha} \boldsymbol{x}^\top \mathcal{L} \boldsymbol{x}$$

$$= \frac{\epsilon}{\alpha} \lambda_1 \tag{38}$$

By combining the above inequality and Equation (35), we have

$$\|\boldsymbol{H}^{(l+1)} - \boldsymbol{H}'^{(l+1)}\|_F = \leq \frac{\epsilon}{\alpha} \lambda_1 \|\boldsymbol{H}^{(l)} \boldsymbol{\Theta}^{(l)}\|_F \tag{39}$$

$\square$

## C   Experimental Settings

### C.1   Software and Hardware Descriptions

Table 4: Package configurations of our experiments.

| Package | Version |
|---|---|
| CUDA | 11.1 |
| pytorch_sparse | 0.6.12 |
| pytorch_scatter | 2.0.8 |
| pytorch_geometric | 1.7.2 |
| pytorch | 1.9.0 |
| OGB | 1.3.1 |

All experiments are conducted on a server with four NVIDIA 3090 GPUs, four AMD EPYC 7282 CPUs, and 252GB host memory. We implement all models based on Pytorch and Pytorch Geometric. During our experiments, we found that **the version of Pytorch, Pytorch Sparse, and Pytorch Scatter can significantly impact the running speed of the baseline.** Here we list the details of our used packages in all experiments in Table 4.

### C.2   Statistics of benchmark datasets

We give the detailed statistics and URLs for all datasets used in our experiments in Table 5. We follow the standard data splits and all datasets are directly downloaded from Pytorch Geometric or the protocol of OGB Hu et al. (2020).

Table 5: Dataset Statistics.

| Dataset | Nodes | Edges | Features | Classes | Label Rates |
|---|---|---|---|---|---|
| Reddit | 232,965 | 11,606,919 | 602 | 41 | 65.86% |
| Yelp | 716,847 | 6,977,409 | 300 | 100 | 75.00% |
| *ogbn-arxiv* | 169,343 | 1,157,799 | 128 | 40 | 53.70% |
| *ogbn-proteins* | 169,343 | 1,157,799 | 128 | 40 | 53.70% |
| *ogbn-products* | 2,449,029 | 61,859,076 | 100 | 47 | 8.03% |

### C.3 Hyperparameter Settings

Regarding Reddit, and Yelp dataset, we follow the hyperparameter configurations reported in the respective papers as closely as possible. Following Fey et al. (2021), we clips the gradient during training. The "Gradient Clipping" in below tables indicate the maximum norm for gradients. "Gradient Clipping= 0.0" means we do not clip the gradients in that experiment. Regarding *ogbn-arxiv* and *ogbn-products* dataset, we follow the hyperparameter configurations and codebases provided on the OGB Hu et al. (2020) leader-board. Please refer to the OGB website for more details. Table 9 summarizes the hyperparameter configuration of GraphSAINT. Table 6, Table 7, and Table 8 summarize the hyperparameter configuration of full-Batch GCN, full-Batch GraphSAGE, and full-batch GCNII, respectively.

Table 6: Configuration of Full-Batch GCN.

| Dataset | Training | | | | Archtecture | | |
|---|---|---|---|---|---|---|---|
| | Learning Rates | Epochs | Dropout | Gradient Clipping | BatchNorm | Layers | Hidden Dimension |
| Reddit | 0.01 | 400 | 0.5 | 0.5 | Yes | 2 | 256 |
| Yelp | 0.01 | 500 | 0.1 | 0.5 | Yes | 2 | 512 |
| *ogbn-proteins* | 0.01 | 1000 | 0.5 | 0.0 | No | 3 | 256 |
| *ogbn-arxiv* | 0.01 | 500 | 0.5 | 0.5 | Yes | 3 | 128 |

Table 7: Configuration of Full-Batch GraphSAGE.

| Dataset | Training | | | | Archtecture | | |
|---|---|---|---|---|---|---|---|
| | Learning Rates | Epochs | Dropout | Gradient Clipping | BatchNorm | Layers | Hidden Dimension |
| Reddit | 0.01 | 400 | 0.5 | 0.5 | Yes | 2 | 256 |
| Yelp | 0.01 | 500 | 0.1 | 0.5 | Yes | 2 | 512 |
| *ogbn-arxiv* | 0.01 | 500 | 0.5 | 0.5 | Yes | 3 | 128 |
| *ogbn-proteins* | 0.01 | 1000 | 0.5 | 0.0 | No | 3 | 256 |
| *ogbn-products* | 0.002 | 500 | 0.5 | 0.5 | No | 3 | 256 |

Table 8: Configuration of Full-Batch GCNII.

| Dataset | Training | | | | Archtecture | | |
|---|---|---|---|---|---|---|---|
| | Learning Rates | Epochs | Dropout | Gradient Clipping | BatchNorm | Layers | Hidden Dimension |
| Reddit | 0.01 | 400 | 0.5 | 0.5 | Yes | 4 | 256 |
| Yelp | 0.01 | 500 | 0.1 | 0.5 | Yes | 4 | 512 |
| *ogbn-proteins* | 0.01 | 1000 | 0.5 | 0.0 | No | 4 | 256 |
| *ogbn-arxiv* | 0.001 | 1000 | 0.1 | 0.1 | Yes | 16 | 256 |

Table 9: Training configuration of GraphSAINT.

| Dataset | RandomWalk Sampler | | Training | | | | Archtecture | | |
|---|---|---|---|---|---|---|---|---|---|
| | Walk length | Roots | Learning Rates | Epochs | Dropout | Gradient Clipping | BatchNorm | Layers | Hidden Dimension |
| Reddit | 4 | 2000 | 0.01 | 40 | 0.1 | 0.5 | Yes | 2 | 128 |
| Yelp | 2 | 1250 | 0.01 | 75 | 0.1 | 0.5 | Yes | 2 | 512 |
| ogbn-arxiv | 3 | 10000 | 0.01 | 500 | 0.5 | 0.5 | Yes | 3 | 256 |
| ogbn-proteins | 3 | 10000 | 0.01 | 120 | 0.5 | 0.0 | No | 3 | 256 |
| ogbn-products | 3 | 20000 | 0.01 | 20 | 0.5 | 0.0 | No | 3 | 256 |

