# OpenReview forum: "DSpar: An Embarrassingly Simple Strategy for Efficient GNN training and inference via Degree-based Sparsification"
_TMLR — Accepted by TMLR_

### Review · Reviewer_rtCN · 2023-03-14

**Summary Of Contributions:**

This paper presents an algorithm to sparsify the input graph for GNNs in order to speedup the execution speed of model training and inference. The sparsification is operated on the global graph level based on importance edge sampling, where the importance score is inversely proportional to the degree of the end nodes. Though the degree-based sampling is simple, it is justified by spectral graph theory that such approximation preserves the spectrum of the original graph (highest and lowest eigenvalues of the graph Laplacian). The light-weight sparsification algorithm leads to low pre-processing overhead and thus speedup in end-to-end model execution. The overall model accuracy is well preserved on graphs with global clustering structure.


**Audience:**

Yes

**Broader Impact Concerns:**

As far as I can see, there is little negative societal impact or ethical implication.

**Claims And Evidence:**

Yes

**Requested Changes:**

The following changes would be critical for me to consider the paper as above the acceptance threshold:
* The fixes regarding the theoretical derivation of Theorem 2 and (5). Details have been elaborated in the “Weaknesses” section
* From the idea / model design level, an in-depth comparison with the GraphSAINT edge sampler would be great.

Minor issues:
* Fig 4: should the x-axis be “Eigenvalue index” or “Singular value index”?


**Strengths And Weaknesses:**

Strengths

+ The paper is clearly written. The ideas are well motivated, the theoretical analysis is mostly sound, and the empirical evaluation covers different aspects to help readers understand the effect of the algorithm.
+ The proposed method is simple and can be efficiently performed in practice on large scale graphs. The algorithmic parameters are connected to spectral graph theory. The analysis exhibits some level of technical depth.
+ The experiments are performed thoroughly on multiple standard benchmarks and on both training and inference. The performance gains seem reasonably high (1.4 - 2.8x speedup).


Weaknesses

- Issues in theoretical analysis:
    - Proof of Theorem 2: The properties of the $\Pi$ matrix seem a bit problematic. If $\Pi$ is a diagonal matrix and $\Pi \Pi = \Pi$, that means for each diagonal element, we have $\Pi_{e,e}^2 = \Pi_{e,e}$, which means $\Pi_{e,e} = 1$ or $0$. This doesn’t sound correct. Is there a typo?
    - Proof of Theorem 2: How do you derive the last inequality (the bound on $||y||^2_2$) in (16)? When we plug in the expression of (14), it seems there would be some $\sum_e R_e$ term appearing in the $||y||^2_2$ expression (due to normalization of all $p_e$ to 1). How do you proceed to the bound of $|\mathcal{V}|-1$?
    - Derivation of (5) in Appendix B.2: It seems to me that the process to go from (21) to (22) is via assigning $x$ as the k-th eigenvector of the Laplacian $\mathcal{L}$. However, for (22) to hold, wouldn’t it require that $\mathcal{L}$ and $\mathcal{L}’$ have the same eigenvectors? Otherwise, if $x^T\mathcal{L}x=\lambda_i$, we cannot derive $x^T\mathcal{L}’x=\lambda_i$ since some other vectors $x’\neq x$ would correspond to $x’^T\mathcal{L}’x'=\lambda\_i'$.
- While the edge sampling probability $p_e$ is justified via spectral graph theory, the expression of $\frac{1}{d_u}+\frac{1}{d_v}$ also looks exactly the same as the sampling probability defined by the GraphSAINT edge sampler. There may be some deeper connections between DSpar and GraphSAINT in addition to the fact that one does sparsification and the other performs subgraph sampling. The authors should elaborate on this aspect.
- The error due to the degree based sparsification highly depends on the graph structure. The theoretical bound on error / sampling complexity may not be practically good if the graph does not exhibit clear clustering structures.

---

> ### Author Response · Authors · 2023-04-16
> **Response to Reviewer rtCN**
>
>
> We thank you for your constructive feedback of our method.  We are glad that you recognized the motivation, soundness,  and effectiveness of our work. We have revised the paper accordingly and highlighted the revision to address your concerns with ${\color{purple}{purple}}$ color.
>
>
>
> ### Requested Changes
>
> **1. The fixes regarding the theoretical derivation of Theorem 2 and (5). Details have been elaborated in the “Weaknesses” section**
>
> We appreciate your valuable feedback and have made the necessary corrections.
>
> * Regarding Theorem 2, you are right, $\Pi$ is a projection matrix (i.e., $\Pi^2=\Pi$) instead of diagnoral matrix. We apologize that this is a typo. Additionally, we have included more context for the proof of Theorem 2, and we apologize for neglecting to mention $\sum R_e=|\mathcal{V}|-1$ in the previous version.
>
> * Regarding Equation 5, we apologize for the lack of details in the previous version. We align $\lambda_i$ with $\lambda_i'$ via Courant-Fischer Theorem. We provide down-to-ground proof for Equation 5.
>
> **2. From the idea / model design level, an in-depth comparison with the GraphSAINT edge sampler would be great.**
>
> Thank you for this suggestion. We add the comparision with GraphSAINT in Section 4.1 and provided a summary below for your convenience. Here we would like to highlight three key difference between them.
>
> * First, they generate subgraph differently.
>     The edge sampler in GraphSAINT is used to build *node-induced subgraph*. Namely, it first select a subset of **anchor nodes** using the edge sampler and including all the edges that connect those nodes. In other words, the induced subgraph may contain edges that are not sampled. In contrast, the graph sparsified by DSpar can be viewed as **edge-induced subgraph**. Namely, DSpar first select a subset of edges from the original graph and including only those nodes that are endpoints of the selected edges.
>
> * Second, they are executed differently. One key difference between DSpar and the graph sampler (e.g., edge sampler and FastGCN sampler) is that **DSpar only sparsify the graph once before training**. In contrast, the graph sampler generates different subgraphs at each training steps.
>
> * Third, they are derived differently, and thus are used differently. $\frac{1}{d_u}+ \frac{1}{d_v}$ in GraphSAINT edge sampler is obtained by debiasing node embeddings in the sampled subgraph. We note that this bias term stems from using the node-induced subgraph. In contrast, DSpar is derived following the traditional spectral graph theory, which mainly focus on providing one light-weight sketch of the original graph.

---

> > ### Comment · Reviewer_rtCN · 2023-05-24
> > **Reviewer's response**
> >
> > Thanks for providing the detailed rebuttal. I've read the revision carefully and I think most of my concerns have been addressed. I thus recommend accepting this paper.
> >
> > Some minor issues:
> > * Description on GraphSAINT in the revision: the edge sampling probability is actually derived based on variance minimization. The bias introduced by the subgraph sampler is addressed by an additional normalization term.
> > * Typo in the revised proof: "definiation" -> "definition". Also Theorem 2 is derived based on unweighted graph (e.g., in the new equation added below Eq 15, we assume $A_e = 1$). This should be made clear in the Theorem statement.

---

> > > ### Author Response · Authors · 2023-06-21
> > > **To reviewer rtCN**
> > >
> > > Dear reviewer,
> > >
> > > We are grateful for your time in thoroughly reviewing the revisions to our paper, and for recommending its acceptance. We appreciate the constructive feedback and we will address the mentioned issues in the updated version.

---

### Review · Reviewer_VvH3 · 2023-04-10

**Summary Of Contributions:**

This paper borrows the idea from an existing work (Spielman & Srivastave 2011) to design a graph sparsification algorithm, DSpar, to sub-sample the edges based on the degree information before training. This paper also proves the theoretical guarantees for using DSpar in reserving the expressiveness of the node representations on the graph.  The numerical experiments indicate that the proposed algorithm can attain up to 5.9 times faster than the baseline while maintaining comparable test accuracy. Moreover, during the inference stage, the algorithm can reduce 90% of the latency.

**Audience:**

Yes

**Broader Impact Concerns:**

I do not see any concerns about the ethical implication of this work.

**Claims And Evidence:**

Yes

**Requested Changes:**

1. Please add corresponding experiments to justify current theoretical results, such as plotting $||H^{(l+1)}-H'^{(l+1)}||_F$ against $\epsilon$, $\lambda_1$, or $\alpha$ to validate eqn. (6).

2. A brief explanation of the technical challenges in deriving the theoretical results and a proof sketch should be added. Additionally, if required, more theoretical results can be included on generalization analysis or convergence analysis.

3. I recommend that the authors revise Algorithm 1 to avoid creating confusion that this paper simply utilizes an existing algorithm. The current version could lead to such misunderstandings.

4. Please include a comparison with FastGCN, see [3].

Questions:

1. Can the author provide insight into why the citation network and the protein-protein interaction network differ in terms of graph sparsification rate?

2. I would like more clarification regarding Figure 4 and the corresponding outcomes. I am thinking that the singular vector is more significant than the singular value.

**Strengths And Weaknesses:**

Strengths:

1. The authors' algorithms are backed by some theoretical guarantees, and although simple, they are highly effective.

2. The experiments are meticulously conducted and provide comprehensive information for reproducibility.

Weakness"

1. My main concern is that the theoretical support is not strong enough, because there is no analysis of generalization, and the difficulties in deriving the relevant proofs are not adequately explained. 2. Here are some recent works providing generalization analysis for GNN sparsification, see [1] & [2], which may inspire the authors to strengthen their theoretical contributions.

[1] Hongkang Li, et al. "Generalization Guarantee of Training Graph Convolutional Networks with Graph Topology Sampling." In International Conference on Machine Learning, pp. 13014-13051. PMLR, 2022.

[2] Shuai Zhang, et al. "Joint Edge-Model Sparse Learning is Provably Efficient for Graph Neural Networks." In The Eleventh International Conference on Learning Representations.

2. I believe [3] adopts a comparable approach that utilizes degree information to identify crucial nodes, which is similar to the technique outlined in this paper. It would be helpful if the authors could provide a comparison between their method and [3].

[3] Jie Chen, et al. "FastGCN: Fast learning with graph convolutional networks via importance sampling." In International Conference on Learning Representations. International Conference on Learning Representations, ICLR, 2018.

3. This paper lacks numerical experiments to justify the tightness and accuracy of the theoretical bound. As a result, it is challenging to validate the theoretical contributions.

---

> ### Author Response · Authors · 2023-04-16
> **Response to Reviewer VvH3 [1/2]**
>
> We sincerely appreciate your time and effort in reviewing our paper. We are glad that you recognized the simplicity, reproducibility,  and effectiveness of our work. We are also grateful for your constructive suggestions on the presentation and experiments in our paper. We have revised the paper accordingly and highlighted the revision to address your concerns with ${\color{orange}{orange}}$ color.
>
> ### Requested Changes
>
> **1. Add corresponding experiments to justify current theoretical results, such as plotting $||\boldsymbol{H}^{(l+1)} - \boldsymbol{H'^{(l+1)}}||$ against $\epsilon,\lambda_1$, or $\alpha$.**
>
> We appreciate your insightful suggestion. We would like to clarify that calculating the error $|\boldsymbol{H}^{(l+1)} - \boldsymbol{H'^{(l+1)}}|_F$ against $\lambda_1, \alpha$ for real-world graphs is quite challenging, as $\lambda_1, \alpha$ are eigenvalues of the graph Laplacian. For instance, plotting $||\boldsymbol{H}^{(l+1)} - \boldsymbol{H'^{(l+1)}}||_F$ against $\lambda_1/\alpha$ requires examining numerous real-world graphs with distinct $\lambda_1/\alpha$ values (computing the eigenvalues of real-world graphs can be extremely time-consuming).
>
> Thus, we fix the input graph and plot the relative Frobenius norm error $\|\boldsymbol{H}^{(l+1)} - \boldsymbol{H'^{(l+1)}}\|_F/\|\boldsymbol{H}^{(l+1)} \|\|_F$ against the error term $\epsilon$ for Cora dataset. We add this ablation study in Section 3.3 Figure 3.
>
>
> **2. A brief explanation of the technical challenges in deriving the theoretical results and a proof sketch should be added. Additionally, if required, more theoretical results can be included on generalization analysis or convergence analysis.**
>
> We appreciate your feedback and have included proof sketches for Theorem 2 and Theorem 3 in Appendix B.1 and Appendix B.3, respectively. Also we thank you for pointing us to [1, 2], we included this discussion in the Section 4.2 Limitation.
>
> Here we would like to highlight two key points. First, **following prior spectral sparsification research [3], our main theoretical result (Theorem 2) is derived without making any assumptions.** Obtaining a meaningful generalization bound for GNNs without any assumptions is highly challenging. We consider proving the generalization bound of GNNs with additional assumptions to be beyond the scope of our paper. Second, as reflected in the listed contributions, **theoretical results are not the primary focus of this paper**. Our main goal is to build a ready-to-use tool for efficiently sparsifying the graph once before training, with wall-clock time speedup.
>
> Furthermore, we would like to emphasize that the generalization of the model trained on the sparsified graph has been systematically and comprehensively assessed using several real-world large graphs.
>
>
> [1] Generalization Guarantee of Training Graph Convolutional Networks with Graph Topology Sampling
>
> [2] Joint Edge-Model Sparse Learning is Provably Efficient for Graph Neural Networks
>
> [3] Graph Sparsification by Effective Resistances

---

> ### Author Response · Authors · 2023-04-16
> **Response to Reviewer VvH3 [2/2]**
>
> **3. I recommend that the authors revise Algorithm 1 to avoid creating confusion that this paper simply utilizes an existing algorithm. The current version could lead to such misunderstanding.**
>
> We respectively point out that we have explicitly mentioned Algorithm 1 is not our original contribution **at least twice**. For example, (1) we directly cite the source of Algorithm 1 in its title. (2), we explicitly mentioend "*Albeit Algorithm 1 is well-established, up to our knowledge, ...*" in Section 3.1. Moreover, in Section 3.1 we mentioned Spielman & Srivastava (2011) serveral times to give the context on why Algorithm 1 is hard to be applied to large graphs. Our unique contribution focuses on demonstrating how and why we could approximate the effective resistance with only the degree information. Thus, We respectfully disagree with the comment "The current version could lead to such misunderstanding".
>
> **4. Please include a comparison with FastGCN.**
>
> We add the comparision with FastGCN in Section 4.1 and provided a summary below for your convenience.
>
> The FastGCN sampler shares some similarities with our method, as both techniques employ node degree information. However, they utilize this information in significantly different ways.
> Specifically, **for a given batch of nodes, FastGCN samples neighbors for each in-batch node with a probability proportional to the square of the node degree.** In contrast, DSpar uses node degree to subsample edges with a probability proportional to $\frac{1}{d_u} + \frac{1}{d_v}$ for any edge $e$ in the graph.
>
> This difference in sampling strategies leads to distinct outcomes. In FastGCN, a neighbor with a higher degree has a greater chance of being sampled for a given node. Conversely, in the DSpar method, an edge is less likely to be sampled if the degrees of its endpoints are large. Moreover, FastGCN sampler is developed for selecting neighbors for a given node, which cannot be directly extended to the area of graph sparsification.
>
> ### Questions
>
> **Can the author provide insight into why the citation network and the protein-protein interaction network differ in terms of graph sparsification rate?**
>
> Thank you for asking that question. Based on the information provided in Table 1, it appears that the ogbn-protein (protein-protein interaction network) graph is denser than the citation network (ogbn-arxiv). The ogbn-protein graph contains 132K nodes and 39.5M edges, with an average node degree of 13.7. In contrast, the ogbn-arxiv graph has 169K nodes and 1.2M edges, with an average node degree of 597. This suggests that the ogbn-protein graph contains more redundant information than the ogbn-arxiv graph. Thus we achieve much higher graph sparsification rate for ogbn-protein.
>
>
> **I would like more clarification regarding Figure 4 and the corresponding outcomes. I am thinking that the singular vector is more significant than the singular value.**
>
> When it comes to spectral graph theory, we usually analyze the eigenvalues of the graph Laplacian matrix instead of the eigenvectors because the eigenvalues of the Laplacian matrix are directly related to the properties of the graph, such as its expansion, clustering coefficient, and connectivity [3].
>
> More importantly, eigenvectors are difficult to compute accurately, particularly for large-scale graphs (we usually apply approximated algorithm for large graphs to compute the eigenvectors), whereas eigenvalues can be computed more efficiently and reliably.

---

### Review · Reviewer_t4jW · 2023-04-25

**Summary Of Contributions:**

This paper introduces DSpar, a novel graph sparsification method that approximates effective resistance using only node degree information, significantly reducing ahead-of-training overhead. The authors experimentally and theoretically demonstrate that this approach results in similar node representations on sparsified and non-sparsified graphs. DSpar leads to up to 5.9x faster training and up to 90% reduced latency during inference, while maintaining accuracy, addressing the inefficiencies of running Graph Neural Networks (GNNs) on large graphs.

**Audience:**

Yes

**Claims And Evidence:**

Yes

**Requested Changes:**

1. Is there a benchmark that is more sensitive to edge sparsification? At least with more significant drops with random sparsification at sparsity 94.9%.

2. Compare with other graph sparsification methods in the experiments. Currently only random sparsification is compared. It would be good to understand the difference with other algorithms.

**Strengths And Weaknesses:**

This paper provides a simple method with solid theoretical analysis and experimental results showing that the proposed method accelerates GCN training and inference at almost no cost of accuracy.
However, the main results shown in Table 1 indicate that the tasks evaluated are not very sensitive to sparsification, as even random sampling achieves similar results. In Figure 7, the results of random sparsification only drops by around 3% even at 94.9% sparsity.

---

> ### Author Response · Authors · 2023-04-28
> **Response to Reviewer t4jW**
>
> We sincerely appreciate your time and effort in reviewing our paper. We are delighted that you acknowledged the novelty and effectiveness of our work. To address your comments, we have revised the paper and marked the changes in ${\color{blue}{blue}}$ color for your convenience.
>
>
> ### Requested Changes
> **1. Is there a benchmark that is more sensitive to edge sparsification? At least with more significant drops with random sparsification at sparsity 94.9%.**
>
> Thank you for raising this question. Random sparsification is **indeed a strong baseline for well-connected graph**. We note that the dataset adopted in Figure 7 is Reddit, which is a dense and well-connected graph. To provide a clearer answer to your question, we have conducted sparsification experiments on the ogbn-products dataset, which is the largest sparse graph commonly utilized in this field, at varying sparsity levels. The updated version now includes this experiment in Figure 8 (marked in blue). For random sparsification, we observed an approximate 10% drop in accuracy at 95% sparsity. And DSPar sigificantly outperforms the random sparsification in this case.
>
>
> **2. Compare with other graph sparsification methods in the experiments. Currently only random sparsification is compared. It would be good to understand the difference with other algorithms.**
>
> Thank you for raising this question. As mentioned in Section "Related Works and Discussion," our choice to compare our approach solely to random sparsification is due to the inability of other sparsification methods (i.e., learning-based sparsification and effective-resistance-based sparsification) to scale effectively for large graphs. Furthermore, when dealing with smaller graphs, sparsification may not provide substantial benefits, as there is little need to sacrifice accuracy for efficiency if it is not a primary constraint. Here we provide a summary below for the learning-based sparsification for your convenience:
>
> "these learning-based sparsification methods have extra training process, and thus introduces significant ahead-of-training overhead. Moreover, learning-based methods are not scalable since it need to assign each edge an extra trainable mask variable, which is extremely expensive for large graphs."
>
> To the best of our knowledge, these are the primary sparsification methods available. If you are aware of any others, we would be more than happy to discuss and consider them in our research.

---

> > ### Comment · Reviewer_t4jW · 2023-05-26
> > **Review of Paper865 by Reviewer t4jW**
> >
> > Thanks for the response. I have read it and made my recommendations. Thank you for providing Figure 8. I still hope you can provide comparisons with other methods, even on smaller graphs, and show how intractable they are on larger graphs.

---

> > > ### Author Response · Authors · 2023-06-21
> > > **To reviewer t4jW**
> > >
> > > Dear reviewer,
> > >
> > > We sincerely appreciate your time and effort in going through our response and providing further valuable recommendations.
> > >
> > > Bests,
> > >
> > > Authors

---

### Decision · Action_Editors · 2023-06-19

**Recommendation:** Accept with minor revision

**Comment:**

There are a few additional issues that the authors should consider addressing before publication:
* Figure 2 is a bad visualization of a distribution - showing some kind of histogram would be a lot more informative
* Table 2 only shows sparsification rate and did not show latency improvement numbers (but the text refers to this table for latency improvement)
* In Section 3.3 it would be good to clarify that this is evaluating a single GNN on 2 graphs, the original vs the sparsified, rather than also training on these two types of graphs.
* Claiming “xxx% improvement” without quantifying the benchmark / model type is a bit misleading, I would suggest being scientific about it and avoid over-selling.


**Audience:**

Researchers and practitioners interested in GNNs on large scale graphs should be interested in this paper.  The proposed algorithm is very simple, with good theoretical properties and decent practical gains and has the potential to be adopted in a range of applications.

**Claims And Evidence:**

This paper proposes a new graph sparsification algorithm DSpar based purely on node degrees.  They claim that DSpar has:
* nice theoretical properties and GNNs could learn expressive node representations on graphs sparsified by DSpar
* nice practical improvement in training & inference speed
* decent ready-to-use implementation already available for existing graph learning frameworks including Pytorch Geometric etc.

The paper presents good evidence for all 3 claims.
* The theoretical analysis of DSpar is solid, motivating DSpar from the perspective of approximating the established “effective resistance” quantity, and showing that the degree-based $p_e$ bounds the effective resistance from both above and below (with different coefficients).  The paper also shows that the DSpar sparsified graph approximates the graph Laplacian in quadratic form bounded by a certain error level, and therefore also bounds the eigenvalues of the Laplacian.  Furthermore, the approximation error of the GNN representations computed on the original graph and on the sparsified graph is also bounded, showing good representation learning properties.
* The practical gains in terms of training throughput and inference latency are demonstrated on a range of large scale graph learning tasks using a few commonly used GNN architectures.  The actual gains largely depend on the model architecture and the dataset, but the authors managed to get improved training throughput on all datasets with all model types, with the gain on average being 1~2x.  The latency improvement for inference is reported to be in the range of 30%-90%.
* The code is open sourced and the interested readers can already take a look at the implementation.